# Understanding Linearity of Cross-Lingual Word Embedding Mappings

**Xutan Peng** [†]                                          *x.peng@shef.ac.uk*

**Mark Stevenson** [†]                                *mark.stevenson@shef.ac.uk*

**Chenghua Lin** [†]                                          *c.lin@shef.ac.uk*

**Chen Li** [‡]                                          *palchenli@tencent.com*

[†]*Department of Computer Science, The University of Sheffield*
[‡]*Applied Research Center, Tencent PCG*

**Reviewed on OpenReview:** *https: // openreview. net/ forum? id= 8HuyXvbvqX*

## Abstract

The technique of Cross-Lingual Word Embedding (CLWE) plays a fundamental role in tackling Natural Language Processing challenges for low-resource languages. Its dominant approaches assumed that the relationship between embeddings could be represented by a linear mapping, but there has been no exploration of the conditions under which this assumption holds. Such a research gap becomes very critical recently, as it has been evidenced that relaxing mappings to be non-linear can lead to better performance in some cases. We, for the first time, present a theoretical analysis that identifies the preservation of analogies encoded in monolingual word embeddings as a *necessary and sufficient* condition for the ground-truth CLWE mapping between those embeddings to be linear. On a novel cross-lingual analogy dataset that covers five representative analogy categories for twelve distinct languages, we carry out experiments which provide direct empirical support for our theoretical claim. These results offer additional insight into the observations of other researchers and contribute inspiration for the development of more effective cross-lingual representation learning strategies.

## 1 Introduction

Cross-Lingual Word Embedding (CLWE) methods encode words from two or more languages in a shared high-dimensional space in which vectors representing lexical items with similar meanings (regardless of language) are closely located. Compared with alternative techniques, such as cross-lingual pre-trained language models, CLWE is orders of magnitude more efficient in terms of training corpora[1] and computational power requirements[2]. As a result, the topic has received significant attention as a promising means to support Natural Language Processing (NLP) for low-resource languages (including ancient languages) and has been used for a range of applications, e.g., Machine Translation (Herold et al., 2021), Sentiment Analysis (Sun et al., 2021), Question Answering (Zhou et al., 2021) and Text Summarisation (Peng et al., 2021b).

---

[1]For example, Kim et al. (2020) show that inadequate monolingual data size (fewer than one million *sentences*) is likely to lead to collapsed performance of XLM (Lample & Conneau, 2019) even for etymologically close language pairs. Meanwhile, CLWE can easily align word embeddings for languages such as African Amharic and Tigrinya for which only have millions of *tokens* (Zhang et al., 2020) are available.

[2]For example, XLM-R (Conneau et al., 2020) was trained on 500× Tesla V100 GPUs, whereas the training of VecMap (Artetxe et al., 2018) can be finished on a single Titan Xp GPU.

The most successful CLWE approach, CLWE alignment, learns mappings between independently trained monolingual word vectors with very little, or even no, cross-lingual supervision (Ruder et al., 2019). One of the key challenges of these algorithms is the design of mapping functions. Motivated by the observation that word embeddings for different languages tend to be similar in structure (Mikolov et al., 2013b), many researchers have assumed that the mappings between cross-lingual word vectors are linear (Faruqui & Dyer, 2014; Lample et al., 2018b; Li et al., 2021).

Although models based on this assumption have demonstrated strong performance, it has recently been questioned. Researchers have claimed that the structure of multilingual word embeddings may not always be similar (Søgaard et al., 2018; Dubossarsky et al., 2020; Vulić et al., 2020), which led to the emergence of approaches relaxing the mapping linearity (Glavaš & Vulić, 2020; Wang et al., 2021a) or using non-linear functions (Mohiuddin et al., 2020; Ganesan et al., 2021). These new methods can sometimes outperform the traditional linear counterparts, causing a debate around the suitability, or otherwise, of linear mappings. However, to the best of our knowledge, the majority of previous CLWE work has focused on empirical findings, and there has been no in-depth analysis of the conditions for the linearity assumption.

This paper approaches the problem from a novel perspective by establishing a link between the linearity of CLWE mappings and the preservation of encoded monolingual analogies. Our work is motivated by the observation that word analogies can be solved via the composition of semantics based on vector arithmetic (Mikolov et al., 2013c) and such linguistic regularities might be transferable across languages. More specifically, we notice that if analogies encoded in the embeddings of one language also appear in the embeddings of another, the corresponding multilingual vectors tend to form similar shapes (see Fig. 1), suggesting the CLWE mapping between them should be approximately linear. In other words, we suspect that the preservation of analogy encoding indicates the linearity of CLWE mappings.

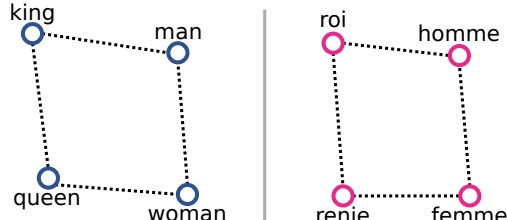

Figure 1: Wiki vectors (see § 4.3) of English (left) and French (right) analogy word pairs based on PCA (Wold et al., 1987). NB: We manually rotate the visualisation to highlight structural similarity.

Our hypothesis is verified both theoretically and empirically. We make a justification that the preservation of analogy encoding should be a *sufficient and necessary* condition for the linearity of CLWE mappings. To provide empirical validation, we first define indicators to qualify the linearity of the ground-truth CLWE mapping ($\mathcal{S}_{\mathrm{LMP}}$) and its preservation of analogy encoding ($\mathcal{S}_{\mathrm{PAE}}$). Next, we build a novel cross-lingual word analogy corpus containing five analogy categories (both semantic and syntactic) for twelve languages that pose pairs of diverse etymological distances. We then benchmark $\mathcal{S}_{\mathrm{LMP}}$ and $\mathcal{S}_{\mathrm{PAE}}$ on three representative series of word embeddings. In all setups tested, we observe a significant correlation between $\mathcal{S}_{\mathrm{LMP}}$ and $\mathcal{S}_{\mathrm{PAE}}$, which provides empirical support for our hypothesis. With this insight, we offer explanations to why the linearity assumption occasionally fails, and consequently, discuss how our research can benefit the development of more effective CLWE algorithms. We also recommend the use of $\mathcal{S}_{\mathrm{PAE}}$ to assess mapping linearity in CLWE applications. We release our data and code at `https://github.com/Pzoom522/xANLG`.

This paper's contributions are summarised as:

- Introduces the previously unnoticed relationship between the linearity of CLWE mappings and the preservation of encoded word analogies.
- Provides a theoretical analysis of this relationship.
- Describes the construction of a novel cross-lingual analogy test set with five categories of word pairs aligned across twelve diverse languages.
- Provides empirical evidence of our claim and introduces $\mathcal{S}_{\mathrm{PAE}}$ to estimate the analogy encoding preservation (and therefore the mapping linearity). We additionally demonstrate that $\mathcal{S}_{\mathrm{PAE}}$ can be used as an indicator of the relationship between monolingual word embeddings, independently of trained CLWEs.

- Discusses implications of these results, regarding the interpretation of previous results and as well as the future development of cross-lingual representations.

## 2  Related Work

**Linearity of CLWE Mapping.**  Mikolov et al. (2013b) discovered that the vectors of word translations exhibit similar structures across different languages. Researchers made use of this by assuming that mappings between multilingual embeddings could be modelled using simple linear transformations. This framework turned out to be effective in numerous studies which demonstrated that linear mappings are able to produce accurate CLWEs with weak or even no supervision (Artetxe et al., 2017; Lample et al., 2018b; Artetxe et al., 2018; Wang et al., 2020; Li et al., 2021).

One way in which this is achieved is through the application of a normalisation technique called "mean centring", which (for each language) subtracts the average monolingual word vector from all word embeddings, so that this mean vector becomes the origin of the vector space (Xing et al., 2015; Artetxe et al., 2016; Ruder et al., 2019). This step has the effect of simplifying the mapping from being *affine* (i.e., equivalent to a shifting operation plus a linear mapping) to *linear* by removing the shifting operation.

However, recent work has cast doubt on this linearity assumption, leading researchers to experiment with the use of non-linear mappings. Nakashole & Flauger (2018) and Wang et al. (2021a) pointed out that structural similarities may only hold across particular regions of the embedding spaces rather than over their entirety. Søgaard et al. (2018) examined word vectors trained using different corpora, models and hyper-parameters, and concluded configuration dissimilarity between the monolingual embeddings breaks the assumption that the mapping between them is linear. Patra et al. (2019) investigated various language pairs and discovered that a higher etymological distance is associated with degraded the linearity of CLWE mappings. Vulić et al. (2020) additionally argued that factors such as limited monolingual resources may also weaken the linearity assumption.

These findings motivated work on designing non-linear mapping functions in an effort to improve CLWE performance. For example, Nakashole (2018) and Wang et al. (2021a) relaxed the linearity assumption by combining multiple linear CLWE mappings; Patra et al. (2019) developed a semi-supervised model that loosened the linearity restriction; Lubin et al. (2019) attempted to reduce the dissimilarity between multilingual embedding manifolds by refining learnt dictionaries; Glavaš & Vulić (2020) first trained a globally optimal linear mapping, then adjusted vector positions to achieve better accuracy; Mohiuddin et al. (2020) used two independently pre-trained auto-encoders to introduce non-linearity to CLWE mappings; Ganesan et al. (2021) obtained inspirations via the back translation paradigm, hence framing CLWE training as to explicitly solve a non-linear and bijective transformation between multilingual word embeddings. Despite these non-linear mappings outperforming their linear counterparts in many setups, in some settings the linear mappings still seem more successful, e.g., the alignment between Portuguese and English word embeddings in Ganesan et al. (2021). Moreover, training non-linear mappings is typically more complex and thus requires more computational resources.

Albeit at the significant recent attention to this problem by the research community, it is still unclear under what condition the linearity of CLWE mappings holds. This paper makes the first attempt to close this research gap by providing both theoretical and empirical contributions.

**Analogy Encoding.**  Analogy is a fundamental concept within cognitive science (Gentner, 1983) that has received significant focus from the NLP community, since the observation that it can be represented using word embeddings and vector arithmetic (Mikolov et al., 2013c). A popular example based on the analogy "*king is to man as queen is to woman*" shows that the vectors representing the four terms ($x_{king}$, $x_{man}$, $x_{queen}$ and $x_{woman}$) exhibit the following relation:

$$x_{king} - x_{man} \approx x_{queen} - x_{woman}. \tag{1}$$

Since this discovery, the task of analogy completion has commonly been employed to evaluate the quality of pre-trained word embeddings (Mikolov et al., 2013c; Pennington et al., 2014; Levy & Goldberg, 2014a). This

line of research has directly benefited downstream applications (e.g., representation bias removal (Prade & Richard, 2021)) and other relevant domains (e.g., automatic knowledge graph construction (Wang et al., 2021b)). Theoretical analysis has demonstrated a link between embeddings' analogy encoding and the Pointwise Mutual Information of the training corpus (Arora et al., 2016; Gittens et al., 2017; Allen & Hospedales, 2019; Ethayarajh et al., 2019; Fournier & Dunbar, 2021). Nonetheless, as far as we are aware, the connection between the preservation of analogy encoding and the linearity of CLWE mappings has not been previously investigated.

## 3   Theoretical Basis

We denote a ground-truth CLWE mapping as $\mathcal{M} : \mathbf{X} \to \mathbf{Y}$, where $\mathbf{X}$ and $\mathbf{Y}$ are monolingual word embeddings independently trained for languages $L_X$ and $L_Y$, respectively.

***Proposition.*** Encoded analogies are preserved during the CLWE mapping $\mathcal{M} \iff \mathcal{M}$ is affine.

***Remarks.*** Following Eq. (1), the preservation of analogy encoding under a mapping can be formalised as

$$\boldsymbol{x_\alpha} - \boldsymbol{x_\beta} = \boldsymbol{x_\gamma} - \boldsymbol{x_\theta} \implies \mathcal{M}(\boldsymbol{x_\alpha}) - \mathcal{M}(\boldsymbol{x_\beta}) = \mathcal{M}(\boldsymbol{x_\gamma}) - \mathcal{M}(\boldsymbol{x_\theta}), \tag{2}$$

where $\boldsymbol{x_\alpha}, \boldsymbol{x_\beta}, \boldsymbol{x_\gamma}, \boldsymbol{x_\theta} \in \mathbf{X}$.

If $\mathcal{M}$ is affine, for $d$-dimensional monolingual embeddings $X$ we have

$$\mathcal{M}(\boldsymbol{x}) \coloneqq M\boldsymbol{x} + \boldsymbol{b}, \tag{3}$$

where $x \in X$, $M \in \mathbb{R}^{d \times d}$, and $\boldsymbol{b} \in \mathbb{R}^{d \times 1}$.

***Proof:*** **Eq. (2)** $\implies$ **Eq. (3).** To begin with, by adopting the mean centring operation in § 2, we shift the coordinates of the space of $\mathbf{X}$, ensuring

$$\mathcal{M}(\vec{0}) = \vec{0}. \tag{4}$$

This step greatly simplifies the derivations afterwards, because from now on we just need to demonstrate that $\mathcal{M}$ is a *linear mapping*, i.e., it can be written as $M\boldsymbol{x}$. By definition, this is equivalent to showing that $\mathcal{M}$ preserves both the operations of addition (a.k.a. additivity) and scalar multiplication (a.k.a. homogeneity).

**Additivity** can be proved by observing that $(\boldsymbol{x_i} + \boldsymbol{x_j}) - \boldsymbol{x_j} = \boldsymbol{x_i} - \vec{0}$ and therefore,

$$(\boldsymbol{x_i} + \boldsymbol{x_j}) - \boldsymbol{x_j} = \boldsymbol{x_i} - \vec{0} \xRightarrow{\text{Eq. (2)}} \mathcal{M}(\boldsymbol{x_i} + \boldsymbol{x_j}) - \mathcal{M}(\boldsymbol{x_j}) = \mathcal{M}(\boldsymbol{x_i}) - \mathcal{M}(\vec{0})$$

$$\xRightarrow{\text{Eq. (4)}} \mathcal{M}(\boldsymbol{x_i} + \boldsymbol{x_j}) = \mathcal{M}(\boldsymbol{x_i}) + \mathcal{M}(\boldsymbol{x_j}). \tag{5}$$

**Homogeneity** can be proved in four steps.

• **Step 1**: Observe that $\vec{0} - \boldsymbol{x_i} = -\boldsymbol{x_i} - \vec{0}$, similar to Eq. (5) we can show that

$$\vec{0} - \boldsymbol{x_i} = -\boldsymbol{x_i} - \vec{0} \xRightarrow{\text{Eq. (2)}} \mathcal{M}(\vec{0}) - \mathcal{M}(\boldsymbol{x_i}) = \mathcal{M}(-\boldsymbol{x_i}) - \mathcal{M}(\vec{0})$$

$$\xRightarrow[\times(-1)]{\text{Eq. (4)}} \mathcal{M}(\boldsymbol{x_i}) = -\mathcal{M}(-\boldsymbol{x_i}). \tag{6}$$

• **Step 2**: Using *mathematical induction*, for arbitrary $\boldsymbol{x_i}$, we show that

$$\forall m \in \mathbb{N}^+, \ \mathcal{M}(m\boldsymbol{x_i}) = m\mathcal{M}(\boldsymbol{x_i}) \tag{7}$$

holds, where $\mathbb{N}^+$ is the set of positive natural numbers, as
*Base Case:* Trivially holds when $m = 1$.
*Inductive Step:* Assume the inductive hypothesis that $m = k$ $(k \in \mathbb{N}^+)$, i.e.,

$$\mathcal{M}(k\boldsymbol{x_i}) = k\mathcal{M}(\boldsymbol{x_i}). \tag{8}$$

Then, as required, when $m = k + 1$,

$$\mathcal{M}\big((k+1)\boldsymbol{x_i}\big) \xlongequal{\text{Eq. (5)}} \mathcal{M}(k\boldsymbol{x_i}) + \mathcal{M}(\boldsymbol{x_i}) \xlongequal{\text{Eq. (8)}} k\mathcal{M}(\boldsymbol{x_i}) + \mathcal{M}(\boldsymbol{x_i}) = (k+1)\mathcal{M}(\boldsymbol{x_i}).$$

• **Step 3**: We further justify that

$$\forall n \in \mathbb{N}^+, \ \mathcal{M}(\frac{\boldsymbol{x_i}}{n}) = \frac{\mathcal{M}(\boldsymbol{x_i})}{n}, \tag{9}$$

which, due to Eq. (4), trivially holds when $n = 1$; as for $n > 1$,

$$\mathcal{M}(\frac{\boldsymbol{x_i}}{n}) = \mathcal{M}\big(\boldsymbol{x_i} + (-\frac{n-1}{n}\boldsymbol{x_i})\big) \xlongequal{\text{Eq. (5)}} \mathcal{M}(\boldsymbol{x_i}) + \mathcal{M}(-\frac{n-1}{n}\boldsymbol{x_i})$$

$$\xlongequal{\text{Eq. (6)}} \mathcal{M}(\boldsymbol{x_i}) - \mathcal{M}(\frac{n-1}{n}\boldsymbol{x_i}) \xlongequal{\text{Eq. (7)}} \mathcal{M}(\boldsymbol{x_i}) - (n-1)\mathcal{M}(\frac{\boldsymbol{x_i}}{n})$$

directly yields $\mathcal{M}(\frac{\boldsymbol{x_i}}{n}) = \frac{\mathcal{M}(\boldsymbol{x_i})}{n}$, i.e., Eq. (9).

• **Step 4**: Considering the set of rational numbers $\mathbb{Q} = \{0\} \cup \{\pm\frac{m}{n} | \forall m, n\}$, Eqs. (4), (6), (7) and (9) jointly justifies the homogeneity of $\mathcal{M}$ for $\mathbb{Q}$. Because $\mathbb{Q} \subset \mathbb{R}$ is a *dense set*, homogeneity of $\mathcal{M}$ also holds over $\mathbb{R}$, see Kleiber & Pervin (1969).

Finally, combined with the additivity that has been already justified above, linearity of CLWE mapping $\mathcal{M}$ is proved, i.e., Eq. (2) $\implies$ Eq. (3). □

***Proof:* Eq. (3) $\implies$ Eq. (2).** Justifying this direction is quite straightforward:

$$\boldsymbol{x_\alpha} - \boldsymbol{x_\beta} = \boldsymbol{x_\gamma} - \boldsymbol{x_\theta} \implies M\boldsymbol{x_\alpha} - M\boldsymbol{x_\beta} = M\boldsymbol{x_\gamma} - M\boldsymbol{x_\theta}$$

$$\implies M\boldsymbol{x_\alpha} + \boldsymbol{b} - (M\boldsymbol{x_\beta} + \boldsymbol{b}) = M\boldsymbol{x_\gamma} + \boldsymbol{b} - (M\boldsymbol{x_\theta} + \boldsymbol{b})$$

$$\implies \mathcal{M}(\boldsymbol{x_\alpha}) - \mathcal{M}(\boldsymbol{x_\beta}) = \mathcal{M}(\boldsymbol{x_\gamma}) - \mathcal{M}(\boldsymbol{x_\theta}). \qquad \square$$

Summarising the proofs for both the forward and reverse directions, we conclude that the proposition holds.

Please note, the high-level assumption of our derivations is that word embedding spaces can be treated as continuous vector spaces, an assumption commonly adopted in previous work, e.g., Levy & Goldberg (2014b), Hashimoto et al. (2016), Zhang et al. (2018), and Ravfogel et al. (2020). Nevertheless, we argue that the inherent discreteness of word embeddings should not be ignored. The following sections complement this theoretical insight via experiments which confirm the claim holds empirically.

## 4 Experiment

Our experimental protocol assesses the linearity of the mapping between each pair of pre-trained monolingual word embeddings. We also quantify the extent to which this mapping preserves encoded analogies, i.e., satisfies the condition of Eq. (2). We then analyse the correlation between these two indicators. A strong correlation provides evidence to support our theory, and *vice versa*. The indicators used are described in § 4.1. Unfortunately, there are no suitable publicly available corpora for our proposed experiments, so we develop a novel word-level analogy test set that is fully parallel across languages, namely xANLG (see § 4.2). The pre-trained embeddings used for the tests are described in § 4.3.

### 4.1 Indicators

#### 4.1.1 Linearity of CLWE Mapping

Direct measurement of the linearity of a ground-truth CLWE mapping is challenging. One relevant approach is to benchmark the similarity between multilingual word embedding, where the mainstream and state-of-the-art indicators are the so-called spectral-based algorithms (Søgaard et al., 2018; Dubossarsky et al.,

2020). However, such methods assume the number of tested vectors to be much larger than the number of dimensions, which does not apply in our scenario (see § 4.2). Therefore, we choose to evaluate linearity via the goodness-of-fit of the optimal linear CLWE mapping, which is measured as

$$\mathcal{S}_{\text{LMP}} \coloneqq -||M^\star X - Y||_F/r \quad \text{with} \quad M^\star = \arg\min_M ||MX - Y||_F,$$

where $||\cdot||_F$ and $r$ denotes the Frobenius norm and the number of $X$'s rows. To obtain matrices $X$ and $Y$, from $\mathbf{X}$ and $\mathbf{Y}$ respectively, we first retrieve the vectors corresponding to lexicons of a ground-truth $\text{L}_\text{X}$-$\text{L}_\text{Y}$ dictionary and concatenate them into two matrices. More specifically, if two vectors (represented as rows) share the same index in the two matrices (one for each language), their corresponding words form a translation pair, i.e., the rows of these matrices are aligned. "Mean centring" is applied to satisfy Eq. (4). For fair comparisons across different mapping pairs, in each of $X$ and $Y$, rows are standardised by scaling the mean Euclidean norm to 1. Generic Procrustes Analysis (not necessarily orthogonal) (Bookstein, 1992) is applied to find $M^\star$.

Large absolute values of $\mathcal{S}_{\text{LMP}}$ mean that the optimal linear mapping is an accurate model of the true relationship between the embeddings, and *vice versa*. $\mathcal{S}_{\text{LMP}}$ therefore indicates the degree to which CLWE mappings are linear.

### 4.1.2 Preservation of Analogy Encoding

To assess how well analogies are preserved across embeddings, we start by probing how analogies are encoded in the monolingual word embeddings. We use the set-based LRCos, the state-of-the-art analogy mining tool for static word embeddings (Drozd et al., 2016).[3] It provides a score in the range of 0 to 1, indicating the correctness of analogy completion in a single language. For the extension in a cross-lingual setup, we further compute the geometric mean:

$$\mathcal{S}_{\text{PAE}} \coloneqq \sqrt{\text{LRCos}(\mathbf{X}) \times \text{LRCos}(\mathbf{Y})},$$

where $\text{LRCos}(\cdot)$ is the accuracy of analogy completion provided by LRCos for embedding $\mathbf{X}$. To simplify our discussion and analysis from now onward, when performing CLWE mappings, by default we select the monolingual embeddings that best encode analogy, i.e., we restrict $\text{LRCos}(\mathbf{X}) \geq \text{LRCos}(\mathbf{Y})$. $\mathcal{S}_{\text{PAE}} = 1$ indicates all analogies are well encoded in both embeddings, and are preserved by the ground-truth mapping between them. On the other hand, lower $\mathcal{S}_{\text{PAE}}$ values indicate deviation from the condition of Eq. (2).

### 4.1.3 Validity of $\mathcal{S}_{\text{PAE}}$

As an aide, we explore the properties of the $\mathcal{S}_{\text{PAE}}$ indicator to demonstrate its robustness for the interested reader. The score produced by LRCos is relative to a pre-specified set of *known* analogies. In theory, a low $\text{LRCos}(\mathbf{X})$ score may not reliably indicate that $\mathbf{X}$ does not encode analogies well since there may be other word pairings within that set that produce higher scores. This naturally raises a question: *does $\mathcal{S}_{\text{PAE}}$ really promise the validity as the indicator of analogy encoding preservation?* In other words, it is necessary to investigate whether there exists an *unknown* analogy word set encoded by the tested embeddings to an equal or higher degree. If there is, then $\mathcal{S}_{\text{PAE}}$ may not reflect the preservation of analogy encoding completely, as unmatched analogy test sets may lead to low LRCos scores even for monolingual embeddings that encode analogies well. We demonstrate that the problem can be considered as an optimal transportation task and $\mathcal{S}_{\text{PAE}}$ is guaranteed to be a reliable indicator.

As analysed by Ethayarajh et al. (2019), the degree to which word pairs are encoded as analogies in word embeddings is equivalent to the likelihood that the end points of any two corresponding vector pairs form a high-dimensional coplanar parallelogram. More formally, this task is to identify

$$\mathbf{P}^\star = \arg\min_{\mathbf{P}} \sum_{\boldsymbol{x} \in \mathbf{X}} \mathcal{C}\big(\mathcal{T}_{\square}^{\mathbf{P}}(\boldsymbol{x})\big), \tag{10}$$

---

[3]We have tried alternatives including 3CosAdd (Mikolov et al., 2013a), PairDistance (Levy & Goldberg, 2014a) and 3Cos-Mul (Levy et al., 2015), verifying that they are less accurate than LRCos in most cases. Still, in the experiments they all exhibit similar trends as shown in Tab. 2.

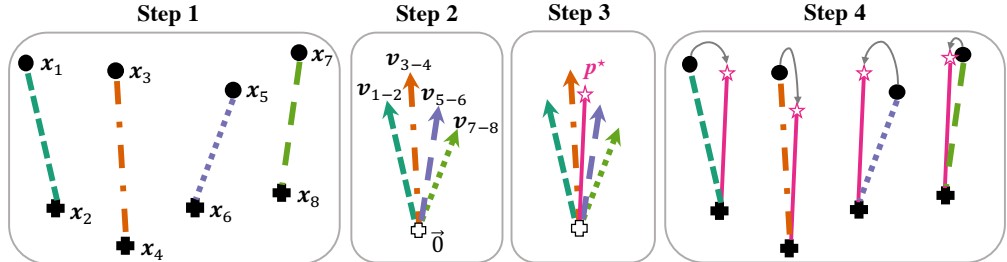

Figure 2: An example of solving $\mathcal{T}^{\mathbf{P}}_{\square}(\cdot)$ in Eq. (11), with $\mathbf{P} = \{(\boldsymbol{x}_1, \boldsymbol{x}_2), (\boldsymbol{x}_3, \boldsymbol{x}_4), (\boldsymbol{x}_5, \boldsymbol{x}_6), (\boldsymbol{x}_7, \boldsymbol{x}_8)\}$. In the figure we adjust the position of $\boldsymbol{x}_1$, $\boldsymbol{x}_3$, $\boldsymbol{x}_5$ and $\boldsymbol{x}_7$ in the last step, but it is worth noting that there also exists other feasible $\mathcal{T}^{\mathbf{P}}_{\square}(\cdot)$ given $\boldsymbol{p}^\star$, e.g., to tune $\boldsymbol{x}_2$, $\boldsymbol{x}_4$, $\boldsymbol{x}_6$ and $\boldsymbol{x}_8$ instead.

where $\mathbf{P}$ is one possible pairing of vectors in $\mathbf{X}$ and $\mathcal{C}(\cdot)$ is the cost of a given transportation scheme. $\mathcal{T}^{\mathbf{P}}_{\square}(\cdot)$ denotes the corresponding cost-optimal process of moving vectors to satisfy

$$\forall \{(\boldsymbol{x}_{\boldsymbol{\alpha}}, \boldsymbol{x}_{\boldsymbol{\beta}}), (\boldsymbol{x}_{\boldsymbol{\gamma}}, \boldsymbol{x}_{\boldsymbol{\theta}})\} \subseteq \mathbf{P},$$
$$\mathcal{T}^{\mathbf{P}}_{\square}(\boldsymbol{x}_{\boldsymbol{\alpha}}) - \mathcal{T}^{\mathbf{P}}_{\square}(\boldsymbol{x}_{\boldsymbol{\beta}}) = \mathcal{T}^{\mathbf{P}}_{\square}(\boldsymbol{x}_{\boldsymbol{\gamma}}) - \mathcal{T}^{\mathbf{P}}_{\square}(\boldsymbol{x}_{\boldsymbol{\theta}}), \tag{11}$$

i.e., the end points of $\mathcal{T}^{\mathbf{P}}_{\square}(\boldsymbol{x}_{\boldsymbol{\alpha}})$, $\mathcal{T}^{\mathbf{P}}_{\square}(\boldsymbol{x}_{\boldsymbol{\beta}})$, $\mathcal{T}^{\mathbf{P}}_{\square}(\boldsymbol{x}_{\boldsymbol{\gamma}})$ and $\mathcal{T}^{\mathbf{P}}_{\square}(\boldsymbol{x}_{\boldsymbol{\theta}})$ form a parallelogram.

Therefore, in each language and analogy category of xANLG, we first randomly sample vector pairing samples, leading to 1e5 different $\mathbf{P}$. Next, for each of them, we need to obtain $\mathcal{T}^{\mathbf{P}}_{\square}(\cdot)$ that minimises $\sum_{\boldsymbol{x} \in \mathbf{X}} \mathcal{C}\big(\mathcal{T}^{\mathbf{P}}_{\square}(\boldsymbol{x})\big)$ in Eq. (10). Our algorithm is explained using the example in Fig. 2, where the cardinality of $\mathbf{X}$ and $\mathbf{P}$ is 8 and 4, respectively.

- **Step 1**: Link the end points of the vectors within each word pair, hence our target is to adjust these end points so that all connecting lines not only have equal length but also remain parallel.
- **Step 2**: For each vector pair $(\boldsymbol{x}_{\boldsymbol{\alpha}}, \boldsymbol{x}_{\boldsymbol{\beta}}) \in \mathbf{P}$, vectorise its connecting line into an offset vector as $\boldsymbol{v}_{\boldsymbol{\alpha}-\boldsymbol{\beta}} = \boldsymbol{x}_{\boldsymbol{\alpha}} - \boldsymbol{x}_{\boldsymbol{\beta}}$.
- **Step 3**: As the start points of all such offset vectors are aggregated at $\vec{0}$, seek a vector $\boldsymbol{p}^\star$ that minimises the total transportation cost between the end point of $\boldsymbol{p}^\star$ and those of all offset vectors (again, note they share a start point at $\vec{0}$).
- **Step 4**: Perform the transportation so that all offset vectors become $\boldsymbol{p}^\star$, i.e.,

$$\forall (\boldsymbol{x}_{\boldsymbol{\alpha}}, \boldsymbol{x}_{\boldsymbol{\beta}}) \in \mathbf{P}, \ \mathcal{T}^{\mathbf{P}}_{\square}(\boldsymbol{x}_{\boldsymbol{\alpha}}) - \mathcal{T}^{\mathbf{P}}_{\square}(\boldsymbol{x}_{\boldsymbol{\beta}}) = \boldsymbol{p}^\star.$$

In this way, the tuned vector pairs can always form perfect parallelograms. Obviously, as $\boldsymbol{p}^\star$ is at the cost-optimal position (see Step 3), this vector-adjustment scheme is also cost-optimal.

Solving $\boldsymbol{p}^\star$ for high dimensions is non-trivial in real world and is a special case of the NP-hard Facility Location Problem (a.k.a. the P-Median Problem) (Kariv & Hakimi, 1979). We, therefore, use the `scipy.optimize.fmin` implementation of the Nelder-Mead simplex algorithm (Nelder & Mead, 1965) to provide a good-enough solution. To reach convergence, with the mean offset vector as the initial guess, we set both the absolute errors in parameter and function value between iterations at 1e4. We experimented with implementing $\mathcal{C}(\cdot)$ using mean Euclidean, Taxicab and Cosine distances respectively. For all analogy categories in all languages, $\mathbf{P}^\star$ coincides perfectly with the pre-defined pairing of xANLG. This analysis provides evidence that the situation where *an unknown kind of analogy is better encoded than the ones used* does not occur in practice. $\mathcal{S}_{\mathrm{PAE}}$ is thus trustworthy.

## 4.2 Datasets

Calculating the correlation between $\mathcal{S}_{\mathrm{LMP}}$ and $\mathcal{S}_{\mathrm{PAE}}$ requires a cross-lingual word analogy dataset. This resource would allow us to simultaneously (1) construct two aligned matrices $X$ and $Y$ to check the linearity of CLWE mappings, and (2) obtain the monolingual LRCos scores of both $\mathbf{X}$ and $\mathbf{Y}$. Three relevant resources were identified, although none of them is suitable for our study.

| | Category | # | 🇩🇪 DE | 🇬🇧 EN | 🇪🇸 ES | 🇫🇷 FR | 🇮🇳 HI | 🇵🇱 PL |
|---|---|---|---|---|---|---|---|---|
| xANLG$_G$ | CAP† | 31 | Budapest Ungarn | Budapest Hungary | Budapest Hungría | Budapest Hongrie | बुडापेस्ट हंगरी | Budapeszt Węgry |
| | GNDR† | 30 | sohn tochter | son daughter | hijo hija | fils fille | बेटा बेटी | syn córka |
| | NATL† | 34 | Peru Peruanisch | Peru Peruvian | Perú Peruano | Pérou Péruvien | पेरू पेरू | Peru Peruwiański |
| | G-PL‡ | 31 | kind kinder | child children | niño niños | enfant enfants | बच्चा बच्चे | dziecko dzieci |
| | Category | # | 🇬🇧 EN | 🇪🇪 ET | 🇫🇮 FI | 🇭🇷 HR | 🇱🇻 LV | 🇷🇺 RU | 🇸🇮 SL |
| xANLG$_M$ | ANIM† | 32 | eagle bird | kotkas lind | kotka lintu | orao ptica | ērglis putns | орёл птица | orel ptica |
| | G-PL‡ | 31 | machine machines | masin masinad | kone koneet | stroj strojevi | mašīna mašīnas | машина машины | stroj stroji |

Table 1: Summary of and examples from the xANLG corpus. # denotes the number of cross-lingual analogy word pairs in each language. †Semantic: animal-species|ANIM, capital-world|CAP, male-female|GNDR, nation-nationality|NATL. ‡Syntactic: grammar-plural|G-PL.

- Brychcín et al. (2019) described a cross-lingual analogy dataset consisting of word pairs from six closely related European languages, but it has never been made publicly available.
- Ulčar et al. (2020) open-sourced the MCIWAD dataset for nine languages, but the analogy words in different languages are not parallel[4].
- Garneau et al. (2021) produced the cross-lingual WiQueen dataset. Unfortunately, a large part of its entries are proper nouns or multi-word terms instead of single-item words, leading to low coverage on the vocabularies of embeddings.

Consequently, we develop xANLG, which we believe to be the first (publicly available) cross-lingual word analogy corpus. For consistency with previous work, xANLG is bootstrapped using established monolingual analogies and cross-lingual dictionaries. xANLG is constructed by starting with a *bilingual* analogy dataset, say, that for $L_X$ and $L_Y$. Within each analogy category, we first translate word pairs of the $L_X$ analogy corpus into $L_Y$, using an available $L_X$-$L_Y$ dictionary. Next, we check if any translation coincides with its original word pair in $L_Y$. If it does, such a word pair (in both $L_X$ and $L_Y$) will be added into the bilingual dataset. This process is repeated for multiple languages to form a cross-lingual corpus.

We use the popular MUSE dictionary (Lample et al., 2018a) which contains a wide range of language pairs. Two existing collections of analogies are utilised:

- **Google Analogy Test Set (GATS)** (Mikolov et al., 2013c), the *de facto* standard benchmark of embedding-based analogy solving. We adopt its extended English version, Bigger Analogy Test Set (BATS) (Gladkova et al., 2016), supplemented with several datasets in other languages inspired by the original GATS: French, Hindi and Polish (Grave et al., 2018), German (Köper et al., 2015) and Spanish (Cardellino, 2019).
- The aforementioned **Multilingual Culture-Independent Word Analogy Datasets (MCIWAD)** (Ulčar et al., 2020).

Due to the differing characteristics of these datasets (e.g., the composition of analogy categories), they are used to produce two separate corpora: xANLG$_G$ and xANLG$_M$. Only categories containing at least 30 word pairs aligned across all languages in the dataset were included. For comparison, 60% of the semantic analogy categories in the commonly used GATS dataset contains fewer than 30 word pairs. The rationale for selecting this value was that it allows a reasonable number of analogy completion questions to be generated.[5] Information in xANLG$_G$ and xANLG$_M$ for the capital-country of Hindi was supplemented with manual

---

[4]Personal communication with the authors.

[5]30 word pairs can be used to generate as many as 3480 unique analogy completion questions such as "*king*:*man* :: *queen*:?" (see Appendix A).

translations by native speakers. In addition, each analogy included in the data set was checked by at least one fluent speaker of the relevant language to ensure that they are valid.

The XANLG dataset contains five distinct analogy categories, including both syntactic (morphological) and semantic analogies, and twelve languages from a diverse range of families (see Tab. 1). From Indo-European languages, one belongs to the Indo-Aryan branch (Hindi|HI), one to the Baltic branch (Latvian|LV), two to the Germanic branch (English|EN, German|DE), two to the Romance branch (French|FR, Spanish|ES) and four to the Slavonic branch (Croatian|HR, Polish|PL, Russian|RU, Slovene|SL). Two non-Indo-European languages, Estonian|ET and Finnish|FI, both from the Finnic branch of the Uralic family, are also included. In total, they form 15 and 21 languages pairs for $\text{XANLG}_\text{G}$ and $\text{XANLG}_\text{M}$, respectively. These pairs span multiple etymological combinations, i.e., intra-language-branch (e.g., ES-FR), inter-language-branch (e.g., DE-RU) and inter-language-family (e.g., HI-ET).

### 4.3  Word Embeddings

To cover the language pairs used in XANLG, we make use of static word embeddings pre-trained on the twelve languages used in the resource. These embeddings consist of three representative open-source series that employ different training corpora, are based on different embedding algorithms, and have different vector dimensions.

- **Wiki**[6]: 300-dimensional, trained on Wikipedia using the Skip-Gram version of FastText (refer to Bojanowski et al. (2017) for details).
- **Crawl**[7]: 300-dimensional, trained on CommonCrawl plus Wikipedia using FastText-CBOW.
- **CoNLL**[8]: 100-dimensional, trained on the CoNLL corpus (without lemmatisation) using Word2Vec (Mikolov et al., 2013c).

## 5  Result

Both Spearman's rank-order ($\rho$) and Pearson product-moment ($r$) correlation coefficients are computed to measure the correlation between $\mathcal{S}_\text{LMP}$ and $\mathcal{S}_\text{PAE}$. Note that, it is not possible to compute the correlations between all pairs due to (1) the number of dimensions varies across embeddings series, and (2) the source and target embeddings have been pre-processed independently for different mappings. Instead, results are grouped by embedding method and analogy category.

Figures in Tab. 2 show that a significant positive correlation between $\mathcal{S}_\text{PAE}$ and $\mathcal{S}_\text{LMP}$ is observed for all setups. In terms of the Spearman's $\rho$, among the 18 groups, 5 exhibit *very strong* correlation ($\rho \geq 0.80$) (with a maximum at 0.96 for CoNLL embeddings on CAP of $\text{XANLG}_\text{G}$), 4 show *strong* correlation ($0.80 > \rho \geq 0.70$), and the others have *moderate* correlation ($0.70 > \rho \geq 0.50$) (with a minimum at 0.58: CoNLL embeddings on ANIM and G-PL of $\text{XANLG}_\text{M}$). Interestingly, although we do not assume a linear relationship in § 3, large values for the Pearson's $r$ are obtained in practice. To be exact, 4 groups indicate very strong correlation, 6 have strong correlation, while others retain moderate correlation (the minimum $r$ value is 0.58: Wiki embeddings on CAP and G-PL of $\text{XANLG}_\text{G}$). These results provide empirical evidence that supplements our theoretical analysis (§ 3) of the relationship between linearity of mappings and analogy preservation.

In addition, we explored whether the analogy type (i.e., semantic or syntactic) affects the correlation. To bootstrap the analysis, for both kinds of correlation coefficients, we divide the 18 experiment groups into two splits, i.e., 12 semantic ones and 6 syntactic ones. After that, we compute a two-treatment ANOVA (Fisher, 1925). For both Spearman's $\rho$ and Pearson's $r$, the results are not significant at $p < 0.1$. Therefore, we conclude that the connection between CLWE mapping linearity and analogy encoding preservation holds across analogy types. We thus recommend testing $\mathcal{S}_\text{PAE}$ *before* implementing CLWE alignment as an indicator of whether a linear transformation is a good approximation of the ground-truth CLWE mapping.

---

[6]`https://fasttext.cc/docs/en/pretrained-vectors.html`
[7]`https://fasttext.cc/docs/en/crawl-vectors.html`
[8]`http://vectors.nlpl.eu/repository/`

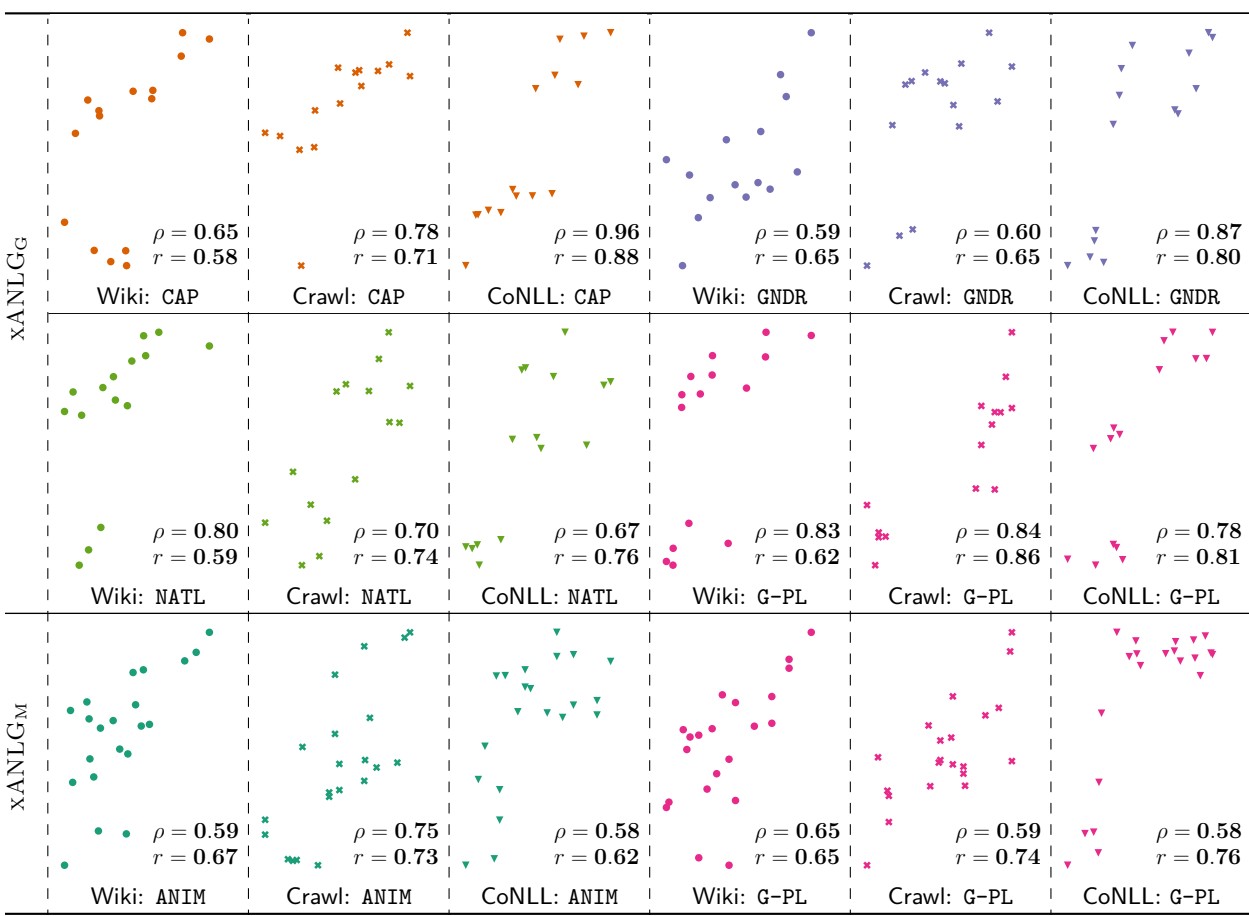

Table 2: Correlation coefficients (Spearman's $\rho$ and Pearson's $r$) between $\mathcal{S}_{\text{LMP}}$ and $\mathcal{S}_{\text{PAE}}$. For all groups, we conduct significance tests to estimate the $p$-value. Empirically, the $p$-value is always less than 1e-2 (in most groups it is even less than 1e-3), indicating a very high confidence level for the experiment results. To facilitate future research and analyses, we present the raw $\mathcal{S}_{\text{LMP}}$ and LRCos data in Appendix B.

Although there are strong correlations between the measures, they are not perfect. We therefore carried out further investigation into the data points in Tab. 2 that do not follow the overall trend. Firstly, we identified that some are associated with "crowded" embedding regions, in which the correct answer to an analogy question is not ranked highest by LRCos but the top candidate is a polysemous term (Rogers et al., 2017). One example is the LRCos score of the CAP analogy for PL's Wiki embeddings, which was underestimated. If we consider the three highest ranked terms, rather than only the top term, then the overall $\rho$ and $r$ of "Wiki: CAP" (the first cell in Tab. 2) will increase sharply to 0.79 and 0.76, respectively.

Secondly, we noticed the in certain cases the source and target vectors of a word pair are too close (i.e. the distance between them is near zero). This phenomenon introduces noise to the results of analogy metrics such as LRCos (Linzen, 2016; Bolukbasi et al., 2016), and consequently, impact $\mathcal{S}_{\text{PAE}}$. For example, the mean cosine distance between G-PL pairs is smaller in $\text{xANLG}_{\text{M}}$ (0.18) than $\text{xANLG}_{\text{G}}$ (0.24). Therefore, the $\mathcal{S}_{\text{PAE}}$ for G-PL is less reliable for $\text{xANLG}_{\text{M}}$ than $\text{xANLG}_{\text{G}}$, leading to a lower correlation.

## 6 Application: Predicting Relationship between Monolingual Word Embeddings

As discussed in § 2, in many scenarios linear CLWE mappings outperform their nonlinear counterparts, while in other setups nonlinear CLWE mappings are more successful. Therefore, an indicator that predicts the relationship between independently pre-trained monolingual word embedding which helps decide whether to

| CLWE method | CCA | | | PROC | | | PROC-B | | DLV | | | RCSLS | | | $\bar{\mathcal{S}}_{\mathrm{PAE}}$ |
|---|---|---|---|---|---|---|---|---|---|---|---|---|---|---|---|
| Seed dict. size | 1K | 3K | 5K | 1K | 3K | 5K | 1K | 3K | 1K | 3K | 5K | 1K | 3K | 5K | |
| EN-FI | .26 | .35 | .38 | .27 | .37 | .40 | .36 | .38 | .27 | .37 | .40 | .31 | .40 | .44 | .41 |
| EN-HR | .22 | .30 | .33 | .23 | .31 | .34 | .30 | .34 | .23 | .31 | .33 | .27 | .36 | .38 | .32 |
| EN-RU | .34 | .43 | .45 | .35 | .45 | .46 | .42 | .45 | .35 | .44 | .47 | .40 | .49 | .51 | .46 |
| FI-HR | .17 | .26 | .29 | .19 | .27 | .29 | .26 | .29 | .18 | .27 | .29 | .21 | .30 | .32 | .23 |
| FI-RU | .21 | .31 | .34 | .23 | .31 | .34 | .32 | .33 | .23 | .31 | .34 | .26 | .34 | .38 | .33 |
| HR-RU | .26 | .35 | .37 | .27 | .35 | .37 | .35 | .37 | .26 | .35 | .37 | .29 | .38 | .40 | .26 |
| Spearman's $\rho$ | **.83** | **.82** | **.86** | **.83** | **.84** | **.88** | **.83** | **.86** | **.84** | **.84** | **.87** | **.87** | **.88** | **.90** | |

Table 3: Spearman's $\rho$ between the Word Translation performance (MRR) of linear-mapping-based CLWE methods (from Glavaš et al. (2019); PROC-B's performance with 5K seed dictionary was not available) and the average analogy encoding preservation score ($\bar{\mathcal{S}}_{\mathrm{PAE}}$).

use linear or non-linear mappings without training actual CLWEs, would be beneficial. Use of this indicator has the potential to reduce the resources required to find optimal CLWEs (e.g., some recent approaches need several hours of processing on modern GPUs (Peng et al., 2021a; Ormazabal et al., 2021)), with corresponding reductions in carbon footprint.

The proposed $\mathcal{S}_{\mathrm{PAE}}$ metric, which can be obtained within several minutes on a single CPU, can be leveraged as such a metric. A high $\mathcal{S}_{\mathrm{PAE}}$ score suggests that the linear assumption holds strongly on the ground-truth CLWE mapping, so it is feasible to train a linear CLWE mapping; otherwise, the non-linear approaches are recommended.

To demonstrate this idea in practice, we revisited a systematic evaluation on CLWE models based on linear mappings (Glavaš et al., 2019), which reported Mean Reciprocal Rank (MRR) of five representative linear-mapping-based CLWE approaches on the Word Translation task (the de facto stadard for CLWEs). We focus on six language pairs (EN-FI, EN-HR, EN-RU, FI-HR, FI-RU, HR-RU) as they are covered by both xANLG$_{\mathrm{M}}$ and the dataset of Glavaš et al. (2019). Additionally, only Wiki embeddings were involved in the experiments of Glavaš et al. (2019). Thus, for each language pair, we aggregated $\mathcal{S}_{\mathrm{PAE}}$ of different analogy categories for Wiki embeddings, then calculated the average, $\bar{\mathcal{S}}_{\mathrm{PAE}}$.

Results are shown in Tab. 3, where the Spearman's $\rho$ between $\bar{\mathcal{S}}_{\mathrm{PAE}}$ and Word Translation performance is highlighted. Strong positive correlations are observed in all setups that were tested. These results demonstrate that $\bar{\mathcal{S}}_{\mathrm{PAE}}$ provides as accurate indication of the real-world performance of linear CLWE mappings, regardless of the language pair, mapping algorithm, or level of supervision (i.e., size of the seed dictionary for training). These results also provide solid support to the main statement of our paper, i.e., the ground-truth CLWE mapping between monolingual word embeddings is linear *iff* analogies encoded in those embeddings are preserved.

# 7 Further Discussion

Prior work relevant to the linearity of CLWE mappings has largely been observational (see § 2). This section sheds new light on these past studies from the novel perspective of word analogies.

**Explaining Non-Linearity.** We provide three suggested reasons why CLWE mappings are sometimes not approximately linear, all linked with the condition of Eq. (2) not being met.

The first may be issues with individual monolingual embeddings (see one such example in the upper part of Fig. 3). In particular, popular word embedding algorithms lack the capacity to ensure semantic continuity over the entire embedding space (Linzen, 2016). Hence, vectors for the analogy words may only exhibit local consistency, with Eq. (2) breaking down for relatively distant regions. This caused the locality of linearity that has been reported by Nakashole & Flauger (2018), Li et al. (2021) and Wang et al. (2021a).

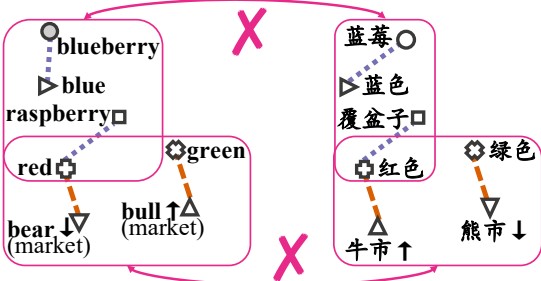

Figure 3: Illustration of example scenarios where the CLWE mapping is non-linear. Translations of English (left) and Chinese (right) terms are indicated by shared symbols. **Upper**: The vector for "*blueberry*" (shadowed) is ill-positioned in the embedding space, so the condition of Eq. (2) is no longer satisfied. **Lower**: In the financial domain some Eastern countries (e.g., China and Japan) traditionally use "*black*" to indicate growth and "*green*" for reduction, while Western countries (e.g., US and UK) assign the opposite meanings to these terms, also not satisfying the condition of Eq. (2).

The second reason why a CLWE mapping may not be linear is semantic gaps. Despite analogies in our xANLG corpus all are language-agnostic, the analogical relations between words may change or even disappear sometimes. For example, languages pairs may have very different grammars, e.g., Chinese does have the plural morphology (Li & Thompson, 1989), so some types of analogy, e.g. G-PL used above, do not hold. Also, analogies may evolve differently across cultures, (see example in the lower part of Fig. 3). These two factors go some way to explain why typologically and etymologically distant language pairs tend to have worse alignment (Ruder et al., 2019).

Thirdly, many studies point out that differences in the domain of training data can influence the similarity between multilingual word embeddings (Søgaard et al., 2018; Artetxe et al., 2018). Besides, we argue that due to polysemy, analogies may change from one domain to another. Under such circumstances, Eq. (2) is violated and the linear assumption no longer holds.

**Mitigating Non-Linearity.** The proposed analogy-inspired framework justifies the success and failure of the linearity assumption for CLWEs. As discussed earlier, it also suggests a method for indirectly assessing the linearity of a CLWE mapping prior to implementation. Moreover, it offers principled methods for designing more effective CLWE methods. The most straightforward idea is to explicitly use Eq. (2) as a training constraint, which has very recently been practised by Garneau et al. (2021)[9]. Based on analogy pairs retrieved from external knowledge bases for different languages, their approach directly learnt to better encode monolingual analogies, particularly those whose vectors are distant in the embedding space. It not only works well on static word embeddings, but also leads to performance gain for large-scale pre-trained cross-lingual language models including the multilingual BERT (Devlin et al., 2019). These results on multiple tasks (e.g., bilingual lexicon induction and cross-lingual sentence retrieval) can be seen as an independent confirmation of this paper's main claim and demonstration of its usefulness.

Our study also suggests another unexplored direction: incorporating analogy-based information into non-linear CLWE mappings. Existing work has already introduced non-linearity to CLWE mappings by applying a variety of techniques including directly training non-linear functions (Mohiuddin et al., 2020), tuning linear mappings for outstanding non-isomorphic instances (Glavaš & Vulić, 2020) and learning multiple linear CLWE mappings instead of a single one (Nakashole, 2018; Wang et al., 2021a) (see § 2). However, there is a lack of theoretical motivation for decisions about how the non-linear mapping should be modelled. Nevertheless, the results presented here suggest that ensembles of linear transformations, covering analogy preserving regions of the embedding space, would make a reasonable approximation of the ground-truth CLWE mappings and that information about analogy preservation could be used to partition embedding

---

[9]They cited our earlier preprint as the primary motivation for their approach.

spaces into multiple regions, between which independent linear mappings can be learnt. We leave this application as our important future work.

## 8 Conclusion and Future Work

This paper makes the first attempt to explore the conditions under which CLWE mappings are linear. Theoretically, we show that this widely-adopted assumption holds *iff* the analogies encoded are preserved across embeddings for different languages. We describe the construction of a novel cross-lingual word analogy dataset for a diverse range of languages and analogy categories and we propose indicators to quantify linearity and analogy preservation. Experiment results on three distinct embedding series firmly support our hypothesis. We also demonstrate how our insight into the connection between linearity and analogy preservation can be used to better understand past observations about the limitations of linear CLWE mappings, particularly when they are ineffective. Our findings regarding the preservation of analogy encoding provide a test that can be applied to determine the likely success of any attempt to create linear mappings between multilingual embeddings. We hope this study can guide future studies in the CLWE field.

Additionally, we plan to expand our theoretical insight to contextual embeddings, inspired by Garneau et al. (2021) who demonstrated that developing mappings that preserve encoded analogies benefits pre-trained cross-lingual language models as well. We also aim to enrich xANLG by including new languages and analogies to enable explorations at an even larger scale. Finally, we will further design CLWE approaches that learn multiple linear mappings between local embedding regions outlined with analogy-based metrics (see § 7).

## Broader Impact Statement

CLWE bridges the gap between languages and is efficient enough to be applied in situations where limited resources are available, including to endangered languages (Zhang et al., 2020; Ngoc Le & Sadat, 2020). This paper presented a theoretical analysis of the mechanisms underlying CLWE techniques which has potential to improve these methods. Moreover, the proposed $\mathcal{S}_{\text{PAE}}$ metric predicts whether monolingual word embeddings in different languages should be aligned using a linear or non-linear mapping, without actually training the CLWEs. This indicator lowers the computational expense required to identify a suitable mapping approach, thereby reducing the computational power needed and negative environmental effects.

Our analysis relies on the use of analogies and previous work has indicated that these may contain biases, e.g., regarding gender (Bolukbasi et al., 2016; Sun et al., 2019). Any future work that incorporates analogies within the CLWE process should be aware of the potential consequences of any biases that may be contained within the analogies used. On the other hand, there is potential for the findings of this work to be leveraged for bias alleviation in cross-lingual representation learning.

## Acknowledgements

We would like to express our sincerest gratitude to all volunteers from Beijing Foreign Studies University who manually annotated and validated the xANLG corpus, as well as Guowei Zhang, Guanyi Chen, Ruizhe Li, Alison Sneyd, and Harish Tayyar Madabushi who helped this study. We also thank the official TMLR reviewers for their insightful comments and Angeliki Lazaridou for the action editing.

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

## A   Question Formulations

For an analogy category with $t$ word pairs, $\binom{t}{2}$ four-item elements can be composed. An arbitrary element, $\alpha{:}\beta :: \gamma{:}\theta$, can yield eight analogy completion questions as follows:

$$\alpha{:}\beta :: \gamma{:}? \quad \beta{:}\alpha :: \theta{:}? \quad \gamma{:}\alpha :: \theta{:}? \quad \theta{:}\beta :: \gamma{:}?$$
$$\alpha{:}\gamma :: \beta{:}? \quad \beta{:}\theta :: \alpha{:}? \quad \gamma{:}\theta :: \alpha{:}? \quad \theta{:}\gamma :: \beta{:}?$$

Hence, $\binom{t}{2} \times 8$ unique questions can be generated.

## B   Raw Data for Tab. 2

| xANLG$_\text{G}$ | | EN-DE | EN-ES | EN-FR | EN-HI | EN-PL | DE-ES | DE-FR | DE-HI | DE-PL | ES-FR | ES-HI | ES-PL | FR-HI | FR-PL | HI-PL |
|---|---|---|---|---|---|---|---|---|---|---|---|---|---|---|---|---|
| Wiki | CAP | .16 | .21 | .17 | .36 | .23 | .21 | .18 | .36 | .22 | .22 | .35 | .25 | .35 | .23 | .33 |
| | GNDR | .32 | .42 | .39 | .26 | .35 | .48 | .40 | .41 | .36 | .39 | .43 | .38 | .30 | .40 | .42 |
| | NATL | .18 | .16 | .15 | .14 | .20 | .19 | .19 | .33 | .21 | .16 | .30 | .21 | .14 | .20 | .32 |
| | G-PL | .22 | .23 | .22 | .36 | .26 | .25 | .23 | .35 | .26 | .25 | .38 | .27 | .37 | .26 | .38 |
| Crawl | CAP | .23 | .23 | .20 | .23 | .29 | .26 | .23 | .24 | .28 | .23 | .26 | .28 | .24 | .29 | .38 |
| | GNDR | .57 | .58 | .59 | .56 | .54 | .65 | .66 | .57 | .59 | .64 | .56 | .57 | .56 | .57 | .58 |
| | NATL | .32 | .43 | .27 | .39 | .29 | .32 | .35 | .47 | .35 | .40 | .43 | .31 | .46 | .31 | .42 |
| | G-PL | .35 | .24 | .33 | .48 | .29 | .33 | .37 | .44 | .42 | .33 | .47 | .33 | .48 | .42 | .51 |
| CoNLL | CAP | .31 | .58 | .32 | .55 | .39 | .58 | .32 | .56 | .38 | .59 | .66 | .59 | .56 | .40 | .55 |
| | GNDR | .48 | .76 | .49 | .55 | .48 | .74 | .55 | .57 | .50 | .77 | .76 | .72 | .59 | .52 | .58 |
| | NATL | .37 | .72 | .26 | .51 | .38 | .78 | .34 | .52 | .36 | .74 | .74 | .73 | .50 | .35 | .50 |
| | G-PL | .32 | .67 | .32 | .48 | .36 | .65 | .34 | .47 | .36 | .68 | .67 | .65 | .50 | .38 | .49 |

| xANLG$_\text{M}$ | | EN ET | EN FI | EN HR | EN LV | EN RU | EN SL | ET FI | ET HR | ET LV | ET RU | ET SL | FI HR | FI LV | FI RU | FI SL | HR LV | HR RU | HR SL | LV RU | LV SL | RU SL |
|---|---|---|---|---|---|---|---|---|---|---|---|---|---|---|---|---|---|---|---|---|---|---|
| Wiki | ANIM | .50 | .50 | .22 | .31 | .19 | .15 | .56 | .27 | .37 | .30 | .35 | .29 | .41 | .30 | .40 | .32 | .36 | .28 | .31 | .22 | .20 |
| | G-PL | .25 | .22 | .37 | .37 | .28 | .33 | .24 | .31 | .29 | .28 | .26 | .30 | .29 | .26 | .27 | .33 | .32 | .30 | .33 | .28 | .28 |
| Crawl | ANIM | .55 | .55 | .55 | .49 | .55 | .51 | .34 | .41 | .45 | .22 | .41 | .40 | .46 | .41 | .45 | .37 | .23 | .28 | .38 | .24 | .43 |
| | G-PL | .28 | .43 | .47 | .43 | .45 | .40 | .30 | .45 | .37 | .43 | .37 | .46 | .40 | .44 | .43 | .42 | .50 | .54 | .39 | .35 | .43 |
| CoNLL | ANIM | .54 | .54 | .99 | .55 | .50 | .53 | .29 | .74 | .46 | .37 | .43 | .87 | .51 | .38 | .46 | .64 | .77 | .98 | .42 | .36 | .41 |
| | G-PL | .45 | .40 | .52 | .42 | .40 | .42 | .37 | .77 | .41 | .41 | .40 | .81 | .37 | .36 | .39 | .84 | .66 | .77 | .36 | .40 | .38 |

Table 4: Raw $\mathcal{S}_\text{LMP}$ results (the negative sign is omitted for brevity).

| | Wiki | | | | Crawl | | | | CoNLL | | | |
|---|---|---|---|---|---|---|---|---|---|---|---|---|
| | CAP | GNDR | NATL | G-PL | CAP | GNDR | NATL | G-PL | CAP | GNDR | NATL | G-PL |
| DE | .68 | .25 | .21 | .23 | .47 | .48 | .79 | .77 | .65 | .43 | .41 | .55 |
| EN | .94 | .33 | .94 | .58 | .57 | .67 | .76 | .94 | .87 | .57 | .79 | .61 |
| ES | .45 | .13 | .35 | .13 | .40 | .57 | .68 | .87 | .13 | .07 | .07 | .17 |
| FR | .92 | .27 | .76 | .13 | .65 | .50 | .85 | .87 | .48 | .14 | .24 | .35 |
| HI | .29 | .30 | .42 | .07 | .58 | .59 | .59 | .32 | .32 | .37 | .31 | .16 |
| PL | .16 | .21 | .26 | .10 | .29 | .55 | .82 | .84 | .45 | .45 | .38 | .52 |

| | Wiki | | Crawl | | CoNLL | |
|---|---|---|---|---|---|---|
| | ANIM | G-PL | ANIM | G-PL | ANIM | G-PL |
| EN | .48 | .65 | .29 | .87 | .36 | .58 |
| ET | .12 | .50 | .52 | 1.00 | .21 | .48 |
| FI | .06 | .65 | .48 | .87 | .42 | .54 |
| HR | .17 | .20 | .50 | .68 | .07 | .11 |
| LV | .19 | .10 | .39 | .84 | .27 | .23 |
| RU | .36 | .40 | .61 | .87 | .42 | .55 |
| SL | .42 | .23 | .39 | .81 | .12 | .39 |

Table 5: Raw monolingual LRCos results (left:xANLG$_\text{G}$; right: xANLG$_\text{M}$).

