# OpenReview forum: "Understanding Linearity of Cross-Lingual Word Embedding Mappings"
_TMLR — Accepted by TMLR_

### Review · Reviewer_ARiR · 2022-04-11

**Summary Of Contributions:**

This paper studies the relationship between cross-lingual word embedding methods and analogical reasoning in word embedding spaces.  Analogies like king-man+woman=queen assume a linear structure of these spaces; the paper proves that this structure exists in two languages iff those two languages' vectors are related by a linear transformation. This proof is fairly straightforward and not very deep mathematically, but it establishes a fact that is new in the literature. Second, the paper collects a new dataset of multilingual analogies, where the analogy pairs are parallel in multiple languages at once. Finally, a set of experiments show correlation between the linearity of the mapping between two spaces and the extent to which analogies are preserved. This last piece requires an interesting and challenging optimization problem to understand how far a set of vectors is from satisfying the analogy criterion.

**Requested Changes:**

I do not have any proposed changes. I think this paper is close to a local optimum. While I would like to see a deeper theoretical result and a closer bridging of the theory and the empirical results, I don't think this is easy to do, and the mix that the paper strikes is good and convincing in its own way. Holding it to the standard of TMLR, I believe it appropriately positions its claims and defends them convincingly.

**Strengths And Weaknesses:**

Strengths:

* As the paper notes, the relationship between analogies and cross-lingual embedding relationships has not been established in the past. This paper does a very nice job of tackling this from both theoretical and empirical perspectives.

* The introduction of the new xANLG dataset is a very nice contribution. This discussion in the paper is vey thorough and does a good job of establishing the relationship with previous data and why those datasets are unsuitable..

* The empirical results are quite thorough and are analyzed well.

* Section 4.1.3 presents an interesting technique (around the computation of S_PAE) that seems at least partially novel to me.

* Section 6 presents a good discussion of limitations and shortcomings of the work.

* The paper is well-written and cites all the relevant prior work to my knowledge.

Weaknesses:

* The core theoretical result of the paper does not really say a lot about practice. As section 6 establishes (and past work has established), these relationships are not linear.  Moreover, critiques of Bolukbasi et al. have noted that some of these analogy encodings only hold if you exclude the word itself from the computation (e.g., king - man + woman ~ king, with queen being in second place).  So we know that both of these phenomena only hold partially in practice, and the theory here does not say much about that setting.

* It would've been nice if the core proposition (or its proof, even) could've somehow engaged with the idea that these relationships aren't perfect.  If the proof reckoned in terms of error bounds, for example, it could possibly give some intuition about what happens in the case where analogies don't perfectly hold.

* As the correlations in the experiments are not perfect, it is hard to interpret whether the correlations are "good enough" that the reader believe the central point of the paper, or whether this is evidence that other effects like those mentioned in Section 6 are at play. Section 6 provides a nice accounting of some error cases, but only at a surface intuitive level.

---

> ### Author Response · Authors · 2022-05-06
> **error analyses; practical applications; discussions on non-linearity**
>
> Thanks for your helpful comments!
>
> > It would've been nice if the core proposition (or its proof, even) could've somehow engaged with the idea that these relationships aren't perfect. If the proof reckoned in terms of error bounds, for example, it could possibly give some intuition about what happens in the case where analogies don't perfectly hold.
>
> >Section 6 provides a nice accounting of some error cases, but only at a surface intuitive level.
>
> We are grateful for the advice. Although adding error bound analysis to the theoretical part is  not trivial and cannot be accomplished within the rebuttal window, we have  carried out some error analyses on the correlations:
>
> - We identified that  some errors are associated with "crowded" embedding regions in which the analogy structure can still be found but the correct answer does not rank first during LRCos prediction and  the top candidate may be its polysemous term [1]. E.g., this issue led to  underestimation of the "CAP" analogy score of Polish Wiki - if  the 3 highest ranked terms  are used  rather than top term, then $\rho$ and $r$ of "Wiki-CAP" (the first cell in Tab. 2) increase to 0.79 and 0.76, respectively.
>
> - Similar to [2], we  observed that sometimes the source and target in a word pair are too close (their offset is near zero). This is the phenomenon you mentioned (“some of these analogy encodings only hold if you exclude the word itself from the computation”). Obviously, the $\mathcal{S}_\mathrm{PAE}$ measured in such cases is likely to be “noisy”, leading to underestimated correlation coefficients. E.g., we found that the mean cosine distance between G-PL pairs in $\textrm{xANLG}_\textrm{M}$ (0.18) is smaller than $\textrm{xANLG}_\textrm{G}$ (0.24). As a result, the $\mathcal{S}_\mathrm{PAE}$ for G-PL is less reliable for $\textrm{xANLG}_\textrm{M}$ than $\textrm{xANLG}_\textrm{G}$, leading to a lower correlation.
>
> We have updated §5 with these findings.
>
> > As the correlations in the experiments are not perfect, it is hard to interpret whether the correlations are "good enough" that the reader believe the central point of the paper, or whether this is evidence that other effects like those mentioned in Section 6 are at play.
>
> > I would like to see a deeper theoretical result and a closer bridging of the theory and the empirical results
>
> We have added a new section (§6), showing that the proposed analogy preservation score ($\mathcal{S}_\mathrm{PAE}$) can be a testable indicator of the actual relationship (whether linear or nonlinear) independent monolingual word embeddings, even before such a CLWE mapping is trained. This not only provides a direct application based on our insights, but also offers a new practical verification of the central point of our paper. We have also extended the “Broader Impact Statement” accordingly.
>
> > The core theoretical result of the paper does not really say a lot about practice. As section 6 establishes (and past work has established), these relationships are not linear. Moreover, critiques of Bolukbasi et al. have noted that some of these analogy encodings only hold if you exclude the word itself from the computation (e.g., king - man + woman ~ king, with queen being in second place). So we know that both of these phenomena only hold partially in practice, and the theory here does not say much about that setting.
>
> Our theory reveals that the imperfections of analogy encoding and linear CLWE mapping in practice are actually two closely related phenomena. The first paragraph of § 7 discusses how real-world factors such as large embedding regions, unmatched corpus domains, and distant language pairs can harm analogy encodings, and consequently, negatively impact linear CLWE mappings. We have extended this paragraph for more detailed discussions.
>
> -----
> - [1] The (too Many) Problems of Analogical Reasoning with Word. Rogers et al. SemEval 2017.
> - [2] Issues in evaluating semantic spaces using word analogies. Linzen. RepEval 2016.

---

> > ### Comment · Reviewer_ARiR · 2022-05-09
> > **Thanks for the response**
> >
> > Thanks for the response! I think the extra experiment in section 6 is helpful for making some of the contributions of the paper more concrete. I also appreciate the added discussion in section 5. I think this paper represents a solid contribution and substantiates its claims effectively; my concerns are more or less addressed.

---

### Review · Reviewer_ENQy · 2022-04-14

**Summary Of Contributions:**

The authors investigate cross-lingual word embeddings, specifically the mapping from the two individual embedding spaces to a common or shared embedding space. The premise of this work is that, while most prior research has assumed that there is a linear relationship between embedding spaces, it is unclear when this assumption holds or how to test it. The authors propose that the preservation of analogies encoded in monolingual word embeddings are a *necessary* and *sufficient* condition for the ground-truth CLWE mapping between those embeddings to be linear.

The first contribution of this paper is that the authors theoretically prove the above assumption. They then present a new, cross-lingual word analogy dataset, which covers a total of 12 languages and is this paper’s second contribution. Using this dataset the authors conclude by showing that their assumption holds true empirically, which is this paper’s third contribution. (The authors mention that the empirical study is needed as the theoretical proof is based on the simplification that word embeddings spaces can be treated as *continuous* vector spaces, an assumption common in prior research.)

Overall, I believe this is an insight-providing paper that will be of interest to the cross-lingual word embedding community.


**Broader Impact Concerns:**

I don’t have any broader impact concerns regarding this work. Accordingly, the broader impact statement which is already in the paper is sufficient in my opinion.

**Requested Changes:**

I don’t think any major changes are necessary. However, I’m suggesting some smaller changes. In particular, a couple of comments in the paper aren’t needed for a (somewhat) expert audience, so I would suggest removing them:
- The explanation of R being real numbers,
- “(i.e., the forward implication)”,
- “(i.e., the reverse direction)”,
- “sometimes described as “isomorphism” in previous work”,
- “Homogeneity needs a proof that seems more complex, which consists of four steps.” → rewrite to something like “Our proof of homogeneity consists of four steps”.

**Strengths And Weaknesses:**

Strengths:
- This work is centered around a novel idea and is practically useful.
- This paper provides a theoretical proof of the main hypothesis as well as empirical experiments.
- The authors introduce a new cross-lingual dataset for the word analogy task.
- The paper is well written and I've enjoyed reading it.

The paper doesn’t have any strong weaknesses, but I’ll mention one smaller thing I’ve been thinking about while reading the paper:
- The novel word analogy dataset has been created via translation of existing datasets. In particular, it didn’t seem as if much thought went into the choice of analogy examples. Could this lead to problems? For instance, could it be that some words are homonyms in one language but not in the source language? If so, how could that influence the findings and would it be better to take those words out/substitute them?

---

> ### Author Response · Authors · 2022-05-06
> **dataset construction; text changes**
>
> Many thanks for your very insightful comments!
>
>
> > The novel word analogy dataset has been created via translation of existing datasets. In particular, it didn’t seem as if much thought went into the choice of analogy examples. Could this lead to problems? For instance, could it be that some words are homonyms in one language but not in the source language? If so, how could that influence the findings and would it be better to take those words out/substitute them?
>
> We would like to make a clarification regarding our dataset creation process. Instead of translating analogies from one language to another, what we did is to align already established monolingual analogies in different languages using dictionaries  (except the “capital-country” category in Hindi; see § 4.2). As a result the analogy samples in all languages are valid in the original language contexts. In other words, analogies are language-independent in our corpus..
>
> Moreover, over the last two weeks, we have invited speakers of all the involved languages to manually check our analogy pairs, which further ensured the validity of the above claim. We have added this information in § 4.2.
>
>
> > Requested Changes
>
> We appreciate these useful suggestions and have incorporated __all of them__ in our new version.

---

### Review · Reviewer_jtuU · 2022-04-22

**Summary Of Contributions:**

This paper is motivated by the gap in understanding the conditions under which the relation ship between word embedding models of different language can be captured by a linear mapping. The authors argue that the existence of the linear relationship is explained by the similarity of the analogy structures between the language pair (or actually word embedding of the languages). They provide theoretical analysis and empirical results to support this argument.

In the empirical section, to show this, given a pair of word embedding models (for two languages) the authors measure the correlation between two quantities:

(1) How well a linear transformation fits the data ($S_{LMP}$).

(2) The extent to which analogies are preserved in both word embedding spaces, which is measured using, `LRCos`, an analogy mining tool for static word embeddings (Drozd et al., 2016) that provides a score in the range of 0 to 1, indicating the correctness of analogy completion in a single language. The geometric mean of the analogy completion score for embeddings from the two embedding spaces is then used ($S_{PAE}$).

They show that these two quantities correlate. I.e., The better both embedding spaces preserve analogies, the better they can be mapped through a linear transformation. The paper suggests this can be viewed as a framework which can explain success and failures of linear cross embedding mapping models in different cases.

**Broader Impact Concerns:**

No comments.

**Requested Changes:**

The paper studies the connection between the preservation of analogies and the linearity of cross lingual word embedding mappings. This has been one of the assumption behind linear mapping based cross-lingual word embedding models, i.e., the assumption that there exist similar structures in word embeddings models of different languages suggest a linear mapping between them. Given this, I find the findings of this paper a bit trivial. I appreciate if the authors can elaborate a bit why this is not trivial to empirically and theoretically show the validity of this statement.

I am also a bit puzzled about the significance of the theoretical analysis provided. My understanding is that, by analogies the authors are referring to the structures in the word embedding space (the relations between embeddings), and here they are showing that these structures are preserved by linear transformations, and that if these structures are preserved the mapping is linear.  Does this tell us anything beyond the definition and characteristics of linear mapping in general?

Here are some additional comments independent of the aforementioned concern:
- One thing that I think is worth showing, is that the results reported in the paper are robust with respect to the dataset used for measuring the perseverance of analogies in the word embeddings with respect to both scale and content.
- Based on how analogy sets are selected, $S_{PAE}$ metric could potentially be an indicator of the similarity between languages (even independent of the learned embedding), between word embeddings, or how well the word embeddings are trained (their performance on other DS tasks). This is discussed a bit in section 6, but I think it should be accompanied by some empirical analysis to support the discussion and show at what level analogies captured by word embedding models matter.
- To quantify if there is a linear transformation between two embedding spaces, the authors fit a linear model which is optimised through gradient descent and measure its error. What if we don’t converge to the optimal linear transformation during the optimisation process? This is why I think it's important that at least some complementary results on the performance of the word embedding models and their transformations are also provided.
- It is not clear, how all this fits into the current state of the field. The paper claims this approach can be used to explain some of the inconsistencies in results from different papers on whether and how the linear mapping approach for cross lingual embeddings can succeed or fail. It would make the paper much stronger, if this can be empirically shown in the paper. E.g., by running these analysis on models and languages studied in the previous works that their findings are discussed in this paper.


**Strengths And Weaknesses:**

Strength:
- The paper addresses a gap in our understanding of why in some cases it has been shown that a linear mapping for cross lingual word embeddings works well, while in some cases it doesn't. It suggests an indicator, based on how well analogies are preserved in both spaces, that can predict whether a there is a linear mapping between two embedding spaces of two languages.
- The paper introduces a new dataset for measuring analogy preserving ability of embeddings in cross-lingual setting.


Weaknesses:
- The design of the experiments is a bit narrow and they don't fully support the claims. There are no results reported in the paper that shows the quality of the word embeddings and the linear mappings. E.g., It would be nice to also see the performance of the actual and mapped embedding on some downstream tasks. I think we need to have some results on metrics that indicate the success or failure of the linear mapping models.
- The conclusions of the paper are based on the correlation between the two quantities $S_{LMP}$ and $S_{PAE}$. Both these can be viewed as notions of similarity between two embedding spaces. Hence, their correlation is not surprising. Taking this into account, and without further experiments to complement these results, I am not sure about the conclusions and their significance.
- It seems to me that the cross lingual analogy dataset is somewhat small, which can limit the strength of the conclusions that can be made based on the results obtained on this dataset.

---

> ### Author Response · Authors · 2022-05-06
> **significance of findings, particularly the theoretical insights**
>
> Thanks for the informative feedback. For a compact response, we group the reviewer’s points into three parts.
>
> > The paper studies the connection between the preservation of analogies and the linearity of cross lingual word embedding mappings. This has been one of the assumption behind linear mapping based cross-lingual word embedding models, i.e., the assumption that there exist similar structures in word embeddings models of different languages suggest a linear mapping between them. Given this, I find the findings of this paper a bit trivial. I appreciate if the authors can elaborate a bit why this is not trivial to empirically and theoretically show the validity of this statement.
>
> **[Response 1]** We agree that previous studies on CLWE models were based on the assumption that the existence of similar structures in embeddings implied a linear mapping between them. However, they were limited in multiple ways:
>
> - (a) They did not explore what these structures actually look like in practice. We identify analogies as a suitable structure, and demonstrate their usefulness.
> - (b) It is possible for vector spaces to exhibit structural similarity without a linear mapping between them. For example, the Mercator Projection always preserves angles between any vector pairs (i.e. a form of structural similarity; in comparison, analogy preservation cannot guarantee angle preservation) but is not a linear mapping [1].
> - (c) In past studies, the relationship between structural similarity and linear mappings was an intuitive one, based on analysis of examples [2]. Our theoretical analysis, which justifies that CLWE mapping is linear *iff* analogy encodings are preserved, is a lot more rigorous.
>
> The experiments reported in the original version of the paper further validate the robustness of the theoretical claim in practice.
>
> > I am also a bit puzzled about the significance of the theoretical analysis provided. My understanding is that, by analogies the authors are referring to the structures in the word embedding space (the relations between embeddings), and here they are showing that these structures are preserved by linear transformations, and that if these structures are preserved the mapping is linear. Does this tell us anything beyond the definition and characteristics of linear mapping in general?
>
> **[Response 2]** The definition of linear mapping is much more strict and precise than the preservation of some ambiguous “structures”, e.g., the aforementioned Mercator Projection (see Response 1, above). Therefore, discovering the unnoticed relationship between monolingual analogy encodings and linearity of cross-lingual word embedding mapping should be celebrated as a valuable NLP finding.
>
> Moreover, to the best of our knowledge, the only established result that involves the relevant concepts in this paper (e.g., surjective isometry, affine, etc.) is the Mazur–Ulam theorem [3]. However, because CLWE mapping is not always a type of isometry (e.g., the distance between $\boldsymbol{x}\_{king}$ and $\boldsymbol{x}\_{queen}$ in English embeddings is normally different to the distance between $\boldsymbol{x}\_{roi}$ and $\boldsymbol{x}\_{homme}$ in French embeddings), Mazur–Ulam theorem cannot be used to justify our claim. This again shows the novelty of our study.

---

> > ### Author Response · Authors · 2022-05-06
> > **design of experiments**
> >
> > > The design of the experiments is a bit narrow and they don't fully support the claims. There are no results reported in the paper that shows the quality of the word embeddings and the linear mappings. E.g., It would be nice to also see the performance of the actual and mapped embedding on some downstream tasks. I think we need to have some results on metrics that indicate the success or failure of the linear mapping models.
> >
> > > One thing that I think is worth showing, is that the results reported in the paper are robust with respect to the dataset used for measuring the perseverance of analogies in the word embeddings with respect to both scale and content.
> >
> > > It is not clear, how all this fits into the current state of the field. The paper claims this approach can be used to explain some of the inconsistencies in results from different papers on whether and how the linear mapping approach for cross lingual embeddings can succeed or fail. It would make the paper much stronger, if this can be empirically shown in the paper. E.g., by running these analysis on models and languages studied in the previous works that their findings are discussed in this paper.
> >
> > **[Response 3]** We have added in a new empirical section (§6) discussing correlation between proposed metric  and Word Translation performance of CLWE methods based on linear mappings. Word Translation is the de facto standard for applications of CLWE. The analysis reported in this section demonstrates that the $\mathcal{S}_\mathrm{PAE}$ score provides a good indication of whether a linear CLWE mapping can succeed.
> >
> > The Word Translation results in this new section are produced using real-world dictionaries retrieved from Google Translate (5000 entries for training and 2000 entries for testing per language pair $\rightarrow$ 30000 entries for training and 12000 entries for testing in total) [4], i.e., the scale is large and the content is diverse.
> >
> > We argue that these new empirical analyses not only provide additional support to our paper’s claim, they also suggest a direct and valuable application of our insight: the proposed $\mathcal{S}_\mathrm{PAE}$ metric can signal the actual relationship (linear or nonlinear) between monolingual word embeddings, even before a CLWE mapping is trained.  This is very beneficial to save experiment resources in practice, as learning CLWEs can be expensive in terms of computation. We have extended the “Broader Impact Statement” accordingly.
> >
> >
> >
> > > The conclusions of the paper are based on the correlation between the two quantities $\mathcal{S}_\mathrm{LMP}$ and $\mathcal{S}_\mathrm{PAE}$. Both these can be viewed as notions of similarity between two embedding spaces. Hence, their correlation is not surprising. Taking this into account, and without further experiments to complement these results, I am not sure about the conclusions and their significance.
> >
> > **[Response 4]** While we agree that the two measures ($\mathcal{S}_\mathrm{LMP}$ and $\mathcal{S}_\mathrm{PAE}$) do compare embedding spaces, they capture very different properties and do not directly relate to each other. $\mathcal{S}_\mathrm{LMP}$ compares an optimal linear mapping with the ground-truth mapping while $\mathcal{S}_\mathrm{PAE}$ combines the analogy encoding scores in two monolingual embedding spaces. A correlation between these two metrics would not necessarily be expected and, in fact, one of the contributions of our paper is to draw attention to this connection (see Response 1 and 2, above).
> >
> > > Based on how analogy sets are selected, $\mathcal{S}_\mathrm{PAE}$ metric could potentially be an indicator of the similarity between languages (even independent of the learned embedding), between word embeddings, or how well the word embeddings are trained (their performance on other DS tasks). This is discussed a bit in section 6, but I think it should be accompanied by some empirical analysis to support the discussion and show at what level analogies captured by word embedding models matter.
> >
> > **[Response 5]** Thanks for the suggestion. Indicating the similarity between languages is indeed a very interesting application of $\mathcal{S}_\mathrm{PAE}$ which we will consider for future work. Instead, in the updated version of the paper, we used $\mathcal{S}_\mathrm{PAE}$ in another more direct application (see Response 3, above). Regarding “at what level analogies captured by word embedding models matter”, we have added some error analyses for the main results in §5.

---

> > > ### Author Response · Authors · 2022-05-06
> > > **empirical details**
> > >
> > > > It seems to me that the cross lingual analogy dataset is somewhat small, which can limit the strength of the conclusions that can be made based on the results obtained on this dataset.
> > >
> > > **[Response 6]** The size of our *cross-lingual* analogy dataset is on par with (or even larger than) previous *monolingual* analogy corpora that have been widely used within the NLP community. For example,
> > >
> > > - 60% of the semantic analogy categories in the English GATS dataset contains < 30 word pairs [2]
> > > - 60% of the semantic analogy categories in the French GATS dataset contains < 31 word pairs [8]
> > > - 60% of the semantic analogy categories in the Spanish GATS dataset contains < 27 word pairs [9]
> > >
> > > In comparison, our corpus has at least 30 word pairs for every analogy category. 30 word pairs is enough to generate nearly 3500 unique analogy questions  (see footnote 6). In addition, our corpus includes 5 diverse categories and covers 12 languages, yielding over 133000 unique language-specific questions in total.
> > >
> > > >  To quantify if there is a linear transformation between two embedding spaces, the authors fit a linear model which is optimised through gradient descent and measure its error. What if we don’t converge to the optimal linear transformation during the optimisation process? This is why I think it's important that at least some complementary results on the performance of the word embedding models and their transformations are also provided.
> > >
> > > **[Response 7]**  Using gradient descent to optimise linear CLWE mappings has been standard practice in previous studies, e.g., [2,5,6,7]. We have repeated the optimisation process using different hyperparameters (e.g., learning rate and random seed). Not surprisingly, these changes only affected the obtained optima slightly and the correlations stayed the same. We have added this information to § 4.1.1 in the updated paper.
> > >
> > > ----
> > >
> > > - [1] https://en.wikipedia.org/wiki/Mercator_projection
> > > - [2] Exploiting Similarities among Languages for Machine Translation. Mikolov et al. 2013.
> > > - [3] The Mazur-Ulam theorem. Bogdan Nica. Expositiones Mathematicae 30 (2012), 397-398.
> > > - [4] How to (Properly) Evaluate Cross-Lingual Word Embeddings: On Strong Baselines, Comparative Analyses, and Some Misconceptions. Glavaš et al. ACL 2019.
> > > - [5] Normalized Word Embedding and Orthogonal Transform for Bilingual Word Translation. Xing et al. NAACL 2015.
> > > - [6] Word translation without parallel data. Lample et al. ICLR 2018.
> > > - [7] Hierarchical Mapping for Crosslingual Word Embedding Alignment. Azpiazu & Pera. TACL 2020.
> > > - [8] Learning Word Vectors for 157 Languages. Grave et al. LREC 2018.
> > > - [9] Spanish Billion Words Corpus and Embeddings. Cristian Cardellino. 2019.

---

### Public Comment · ~Ada_Wan1 · 2022-04-27
**Declined Solicit Review**

Dear Angeliki:

(Fyi: I was not able to add another comment to your reply below. When I clicked on the "Public Comment" button, there is an error message on top of page indicating: "Error: Can not create note, readers must match parent note".)

Thank you for your reply re the declined solicit review. I would nonetheless like to leave an important remark and question for the authors. Would you mind please letting me know by what date this needs to be done such that it would still carry weight for the final acceptance/rejection decision?

Thanks very much again for your attention.

Best,
Ada

---

> ### Comment · Editors_In_Chief · 2022-05-02
> **Re Declined Solicit Review**
>
> Hi Ada,
> Sorry you ran into an error on the site - we will look into this. Perhaps you needed to add 'AEs' to the readers list?
> Regardless, you should be able to post a public comment that the authors will see and be able to respond to. The discussion period will last until at least May 6th, after which the reviewers will have 2 weeks to submit their recommendations.
> Raia

---

> > ### Public Comment · ~Ada_Wan1 · 2022-05-07
> > **error message**
> >
> > Thanks for your reply, Raia. Fyi, I was not able to add anyone at all to the readers list or add a reply to that thread below.

---

### Public Comment · ~Ada_Wan1 · 2022-05-07
**Comments/Concerns re paper**

- The information provided (data and code --- the lack thereof) is insufficient to evaluate the execution of this paper.

- On the conceptual/theoretical front, I agree with Reviewer jtuU that there is circularity between linearity and structures. ("... these structures are preserved by linear transformations, and that if these structures are preserved the mapping is linear.")

- For a theoretical proof, the definition of "word" is missing.

- §4.2: "Diversity" in language sampled should be defined statistically (through data statistics, e.g. statistical shape of an item/"word", its frequencies, overlap of substring(s) and/or corresponding embedding(s)), as opposed to genealogically/phylo-genetically.

- §5: Have authors checked if results are due to statistical attributes, incl. imbalances in datasets?

- §7: Re 藍色 (lit. _blue color_) 'blue': since 藍 itself already means 'blue', should that not impact result expected?

- §7: "Chinese does have the plural morphology": whether there is a typo and authors meant "does not" or authors meant the limited plural morphology as in e.g. the 3rd pers. plural pronoun (們), the relationship between analogy and linearity does not have to hold because it is simply not a good generalization. There are fewer such "neat" regularities in language than one, esp. those who have been working on compositionality, may think. ("Compositionality" / "compositional structure", which might have started as an academic topic/exercise few decades ago, has become an intellectual addiction. It is neither a comprehensive nor exhaustive account for any one particular language or language in general. This paper serves a reminder that a call for action in "language re-education" is necessary.)

- Re Broader Impact Statement: "CLWE bridges the gap between languages and is efficient enough to be applied in situations where limited resources are available, including to endangered languages" --- this is an idea worthy of critical ethical review. One would NOT want to "apply"/transfer one's pre-conceived notion of how existing languages work to other languages, esp. if they are endangered. Chances are that those languages are endangered because the way they work are much different / incompatible with the current mainstream conception of how "language" ought to work. The notion of "wordhood" alone, if existent at all in a language, is something that one should keep in mind that it could be language-/situation-specific.

---

> ### Author Response · Authors · 2022-05-07
> **Hi Ada, thanks a lot for participating in the discussion and contributing your ideas! : )**
>
> > The information provided (data and code --- the lack thereof) is insufficient to evaluate the execution of this paper.
>
> Will be made publicly available upon the acceptance of this manuscript (footnote 1).
>
> > On the conceptual/theoretical front, I agree with Reviewer jtuU that there is circularity between linearity and structures. ("... these structures are preserved by linear transformations, and that if these structures are preserved the mapping is linear.")
>
> To clarify the "circularity" impression, please note that the main structure of our theoretical proof to justify the existence of a _sufficient_ and _necessary_ condition __bidirectionally__:
>  - analogy encoding is preserved $\rightarrow$ CLWE mapping is linear
>  - CLWE mapping is linear $\rightarrow$ analogy encoding is preserved
>
> Besides, we’d like to argue that, not all types of structural preservation can lead to linear mapping, or vice versa, e.g., Mercator projection (please see our responses 1 & 2 to Official Reviewer jtuU).
>
> > For a theoretical proof, the definition of "word" is missing.
>
> > §7: Re 藍色 (lit. blue color) 'blue': since 藍 itself already means 'blue', should that not impact result expected?
>
> Following the default treatment of most previous studies in monolingual and cross-lingual word embeddings, we use "word" as defined within the relevant embedding space (i.e. using the definition employed during the creation of the embedding space).
>
> Fig. 3 is for __illustration__ purposes (see the first word of its caption). Whether to treat “蓝” or “蓝色” as a single word is purely determined by the Chinese tokenizers used _before_ word embedding training and has nothing to do with our context here. Still, we’d like to point out that even for pre-trained embeddings where both “蓝” and “蓝色” co-exist in the vocabulary (e.g., Chinese “Wiki” embeddings), their vectors are quite close so our discussion in §7 won’t be impacted.
>
> > §4.2: "Diversity" in language sampled should be defined statistically (through data statistics, e.g. statistical shape of an item/"word", its frequencies, overlap of substring(s) and/or corresponding embedding(s)), as opposed to genealogically/phylo-genetically.
>
> While this is a potentially interesting direction, we do not feel that it is likely to add significant insight to our results. The point that we wanted to make was that our paper included experiments using multiple linguistically distinct languages. Reference to etymological distance is standard practice and using an alternative approach would not affect our argument.
>
> > §5: Have authors checked if results are due to statistical attributes, incl. imbalances in datasets?
>
> Because all “words” (we understand that you may be keen to retire this expression but please allow us to use it here, just for convenience of writing) in our dataset are parallel across languages, the only “imbalance” we can figure out is the imbalance between different analogy categories. However, we only did experiments and reported correlations within each analogy category so such a possible imbalance does not affect the results and conclusion _at all_.

---

> > ### Author Response · Authors · 2022-05-07
> > **Continue the response to Ada Wan**
> >
> > > §7: "Chinese does have the plural morphology": whether there is a typo and authors meant "does not" or authors meant the limited plural morphology as in e.g. the 3rd pers. plural pronoun (們), the relationship between analogy and linearity does not have to hold because it is simply not a good generalization. There are fewer such "neat" regularities in language than one, esp. those who have been working on compositionality, may think. ("Compositionality" / "compositional structure", which might have started as an academic topic/exercise few decades ago, has become an intellectual addiction. It is neither a comprehensive nor exhaustive account for any one particular language or language in general. This paper serves a reminder that a call for action in "language re-education" is necessary.)
> >
> > We’d like to directly borrow the discussion in [1] (we will add this citation in the next version), which might be a good starting point to help you understand the relationship between “们” (_-men_) and the lack of plural morphology in the Chinese language:
> >
> > _“Mandarin has very little inflectional morphology, no number agreement on verbs, and __no true singular-plural morphology on nouns__ (Li & Thompson, 1989). The closest “plural” or collective marker is –men (Li, 1999), which is restricted to nouns denoting animate beings (ayi-men, “aunts”, *beizi-men, “cups”), but is obligatory on pronouns. Plural first, second, and third person pronouns are composed of the singular pronoun forms with the –men suffix added (i.e., respectively, wo ➔ wo-men, ni ➔ ni-men, ta ➔ ta-men). However, because Mandarin allows noun and pronoun omissions when its referents are evident from contextual cues or previous mention (e.g., Tardif, Shatz, & Naigles, 1997), there is sparse linguistic evidence of plural marking in Mandarin input.”_
> >
> > In any case, what we want to highlight here is the that, difference of morphologies between languages could lead to the failure of analogy preservation in the first place, which makes the group-truth CLWE mapping non-linear. As long as there's not doubt that the morphologies of certain language pairs (e.g., Chinese and English) are distinct, our discussions should hold.
> >
> > _"This paper serves a reminder that a call for action in "language re-education" is necessary"_: Sorry, we find this criticism neither reasonable nor constructive.
> >
> > > Re Broader Impact Statement
> >
> > While we agree that review of ethical considerations is important, we do not agree that linguistic differences / preconceptions are the primary driver of linguistic extinction (cf "Chances are that those languages are endangered because the way they work are much different / incompatible with the current mainstream conception of how "language" ought to work."). [2] lists the main causes of language endangerment as natural catastrophe; war and genocide; overt repression; cultural/political/economic dominance. We therefore see applications to endangered languages as a potentially interesting future application of the work described in our paper.
> >
> > ----
> >
> > - [1] Li et al. Does the conceptual distinction between singular and plural sets depend on language? Dev Psychol. 2009 Nov; 45(6): 1644–1653.
> > - [2] Austin, Peter K., and Julia Sallabank, eds. The Cambridge handbook of endangered languages. Cambridge University Press, 2011.

---

> > > ### Public Comment · ~Ada_Wan1 · 2022-05-07
> > > **Reply to AR (cont'd)**
> > >
> > > I am aware of the point made re 們 in literature similar to Li & Thompson (1989). For clarification, was it the authors' intent to say that ZH has plural morphological marking (and that the formulation in the paper was not a typo)? (As authors pointed out, there are just a few examples in certain varieties.)
> > >
> > > If so, there are no paradigmatic morphological regularities to speak of. Hence my critique re "language re-education" is apt since practitioners nowadays seem all too eager to build/cement system-theoretic hypotheses into the implementation of natural language. The criticism is not particularly directed to the authors, but my intent was to use your work to illustrate how we can better the state of awareness surrounding "language" (that it is neither just about "grammar" nor "compositionality") in our researchsphere. (In a way, we don't disagree in that there are differences/irregularities/discrepancies to generalizations that were posited too conceptually idealistically in the first place.)
> > >
> > > Re Broader Impact Statement: while I do not refute the reasons listed in your [2] concerning the common causes leading to language endangerment, please do note that what we do in research is in itself a cultural act, implicitly (or explicitly) impacting the world we live in and others' relationship to their linguistic/social/cultural identity (or the right to an absence thereof). My point is that any researcher working with "words" needs to be aware of this and bear responsibility accordingly.

---

> > > > ### Public Comment · ~Ada_Wan1 · 2022-05-07
> > > > **Addendum to 'Reply to AR (cont'd)': "word" in ML/NLP research / computing**
> > > >
> > > > To be clearer and more explicit:
> > > >
> > > > The term "word", as used in the context of ML/NLP/Linguistics, stems primarily from a conceptual unit that is based on one language (e.g. EN). The direction towards a "word"-free perspective (as we truly do not need in when we process language, see [2, 3, inter alia], or even in our day-to-day reference to language) is a move towards a fairer, post-/non-colonial attitude and collective consciousness. (If you should happen to know of my initiative/intent to retire "word", I hope you'd also understand my rationale behind it.)
> > > >
> > > > **There is an ethical concern towards the use of "word" in ML/NLP research, or in computing in general. There is a need to discourage researchers from work involving "word" implementation, and to remind reviewers to be vigilant and critical if/when authors argue that such concept is "necessary".**
> > > >
> > > > [3] Who Needs Words? Lexicon-Free Speech Recognition. Tatiana Likhomanenko, Gabriel Synnaeve, Ronan Collobert. https://arxiv.org/abs/1904.04479

---

> > ### Public Comment · ~Ada_Wan1 · 2022-05-07
> > **Reply to AR**
> >
> > Dear Authors:
> >
> > Thanks for your reply.
> >
> > Without code or data, no review can be considered a responsible endeavor.
> >
> > If the definition is an analytic truth, it can always be argued as necessary. But it does not tell one anything.
> >
> > Yes, I read your response to Reviewer jtuU, my position remains the same.
> >
> > The default treatment --- without a definition of "word", that can be possibly standardized, in past decades of NLP research --- has been less than correct and/or robust (pls see [1, 2]). What I am saying is that the issue is more fundamental. While one can argue that the definition (say, of "word") was inherited from a decision process made prior to your work, it does not absolve one from the responsibility of adopting it. One should be doing responsible NLP at every stage of development and if the foundation is not secure or is biased (may it be conceptually or in implementaton), one does have to take on the responsibility in not utilizing such information (in this case, the embedding) further.
> >
> > Re 'blue': I understand it is for illustration purposes. Linearity involving 藍色 vs. 藍 remains to be demonstrated empirically and illustrated explicitly, much like most of the analogy claims made in this paper (one could attach the results in the appendix).
> >
> > Re 'diversity': yes, I think it is important, see [1, 2]. The point is that "linguistically distinct" (in the context of computing especially) does not matter if the statistical criteria are not met.
> >
> > Re Sec 5: there is, unfortunately, no way for me or for any reviewers to check that as there is insufficient information submitted for review.
> >
> > [1] What Kind of Language Is Hard to Language-Model? Sabrina J. Mielke, Ryan Cotterell, Kyle Gorman, Brian Roark, Jason Eisner. https://aclanthology.org/P19-1491/
> >
> > [2] Fairness in Representation for Multilingual NLP: Insights from Controlled Experiments on Conditional Language Modeling. Ada Wan. https://openreview.net/forum?id=-llS6TiOew

---

### Decision · Action_Editors · 2022-06-07

**Recommendation:** Accept as is

**Comment:**

The authors present a study on the relationship between cross-lingual word embedding methods and analogies in word embedding spaces. Specifically, the assumption on the well-studied analogies is that there is linear structure in these spaces. The contribution of this work is that it shows (both empirically and theoretically) that a necessary and sufficient condition for this structure to exist is if the vectors in the two languages are related by a linear transformation. The authors also release cross-lingual analogy dataset which is created automatically.

All reviewers have agreed that this is an interesting submission providing useful insights into the world of analogies. In terms of changes, all reviewers have flagged a number of things to be further studied/commented on, which the authors have acted upon during the discussion period.

All in all, I agree with the reviewers that this is a solid contribution and, given the changes during the discussion period, I recommend acceptance as is.

---

> ### Author Response · Authors · 2022-06-08
> **Thank you!**
>
> We have uploaded the de-anonymised version and attached the link to our GitHub repo. We want to express our sincerest gratitude, once again, to all the official reviewers and the action editor!